# Performance of Asphalt Rubber Mixture Overlays to Mitigate Reflective Cracking

**DOI:** 10.3390/ma15072375

**Published:** 2022-03-23

**Authors:** Liseane Padilha Thives, Jorge C. Pais, Paulo A. A. Pereira, Manuel C. Minhoto, Glicério Trichês

**Affiliations:** 1Department of Civil Engineering, Federal University of Santa Catarina, Florianópolis 88037-000, Brazil; glicerio.triches@ufsc.br; 2Department of Civil Engineering, Campus de Azurém, Universidade do Minho, 48000-058 Guimarães, Portugal; jpais@civil.uminho.pt (J.C.P.); ppereira@civil.uminho.pt (P.A.A.P.); 3Department of Civil Engineering, Instituto Politécnico de Bragança, 5300-253 Bragança, Portugal; minhoto@ipb.pt

**Keywords:** asphalt rubber, reflective cracking, numerical simulation

## Abstract

Adequately predicting overlay behaviour is essential for flexible pavement rehabilitation to reach the predicted lifespan. Reflective cracking is one of the main failure mechanisms affecting overlay performance. This failure may occur due to cracks in the lower layers that propagate to the overlay due to traffic loads, temperature variations, shrinkage cracking of cement-treated layers, and subgrade movements. This work aims to assess the reflective cracking phenomenon of asphalt rubber mixtures as an overlay through laboratory tests and numerical simulation. Four-point bending equipment and the reflective crack device were used to perform the laboratory tests. A numerical simulation through the finite element method was accomplished to estimate the von Mises strain and develop reflective cracking fatigue laws. The results showed that the asphalt rubber mixtures are suitable for extending overlay lifespan considering reflective cracking. The evaluated asphalt rubber mixtures presented reflective cracking resistance almost eight times greater than the conventional ones.

## 1. Introduction

The rehabilitation of distressed flexible pavement surfaces by applying a new asphalt layer (overlay) over the cracked layer is a solution commonly adopted by the Brazilian departments of transportation. However, this practice has proved improper once the new layer has premature failure due to the appearance of cracks propagated from the old ones. Milling application in the existing layer before the overlay is less frequently used in the country.

It is a consensus in the international literature that an overlay does not achieve the expected lifespan without previous treatment of defects of the existing layer. The cracking from the layer to be rehabilitated propagates until it reaches the overlay [1,2,3]. Reflective cracks formed in hot-mix asphalt (HMA) overlays result from horizontal and vertical movements at the joints and cracks of the underlying cracked layers [4].

Many techniques to mitigate reflective cracking in flexible pavement surfaces are available, such as (i) modification or treatment of existing surface (mill and re-place wearing surface, hot in-place recycling; full-depth reclamation); (ii) pre-overlay repairs of existing pavement surface (HMA inlay, HMA patches); (iii) stress/strain relieving interlayer (stress absorption membrane interlayer—SAMI, geosynthetic fabrics); (iv) HMA mixture modification (polymer-modified asphalt, rubberised asphalt, stone matrix asphalt); (v) HMA overlay reinforcement (steel-reinforcing nettings, geotextiles, geogrids); and (iv) crack control (sawing and sealing joints in HMA overlays, chip seal) [4,5,6]. Baek and Al-Quady [6] assert that the reflective cracking mechanism is not yet well understood despite the acquaintance of several techniques to reduce it.

The stress analysis for cracking analysis in an elastic solid was established by Irwin in 1957 [7], and there are three modes in which a crack grows in a structure [8]. Mode I is a tensile mode that corresponds to crack opening and closing movement; Mode II (sliding mode) consists of a shear deformation normal to the crack; and Mode III (tearing mode) represents a shear deformation parallel to the crack.

Paris and Erdogan [9] applied the fracture mechanics approach to evaluate reflective cracking. In the tests, curves were obtained (Appendix A) for a specific number of load cycles (N) corresponding to a crack length (c). The curves can be reduced to the only curve, as represented by Equation (1) [9].
(1)$$N\;=\;\int_{0}^{h}\frac{\mathrm{dc}}{A\;\times \;{\left(\Delta K\right)}^{n}}$$
where N is the number of loading cycles; c is the crack length; ΔK is the stress-intensity factor amplitude, depending on the geometry of the pavement structure, fracture mode, and crack length; A and n are fracture properties of HMA determined by the experimental test.

Three different regions can be distinguished in Appendix A. In Region I, the stress-intensity factor is small enough to prevent no crack growth. Afterwards, stable crack growth is observed (Region II). This region is a straight line represented by Paris and Erdogan’s law [9] (Equation (1)). Region III represents the fatigue life of the material when the stress-intensity factor is equal to the critical stress-intensity factor [8,10,11].

Cracking may appear on flexible pavement surfaces through fatigue, thermal cracking, or reflective mechanisms. Fatigue cracking is due to the accumulated damage produced by the repetition of the loading in load–unload cycles [12]. The existing cracks from the old layer reflect stress concentration zones in overlays.

Under traffic loads and temperature variations, the crack edges are subject to differential movements, producing a stresses concentration that is accountable for the reflective cracks [13,14,15]. The main factors of these differential movements are attributed to traffic action, temperature variations, and subgrade movements [1,16,17,18]. Shrinkage cracking of cement-treated layers also promotes reflexive cracks appearing.

Molenaar and Potter [17] consider traffic actions the principal contributor to reflective cracking appearance. At the same time, temperature variations also lead to changes in HMA stiffness, which results in a tension state in the area above the cracks.

The crack edges movements have different durations depending on the traffic and temperature. Traffic corresponds to rapid cracking movements, and the HMA has elastic mechanical behaviour (almost instantaneous response). In contrast, slow duration solicitations result from thermal retraction and are caused by daily temperature variations, and the HMA tend to have a viscoelastic behaviour with creep or relaxation [19,20].

Moreover, as the truck axles pass over the pavement hundreds of thousands of times per day, the crack movements resulting from traffic have high frequencies. In opposition, the movements caused by daily temperature variations between day and night have low frequencies (twice per day). In the case of seasonal temperature variations, the frequency is low and sometimes occurs once per year [16,18].

The adoption of reflective cracking in pavement rehabilitation design depends on the structure, site temperatures and loading conditions. Thus, the development of laws (fatigue and reflective cracks) are required to describe crack initiation and development phases [10]. In the initiation phase, the predicted lifespan can be determined by calculating the tensile strain at the base of the overlay [12,21,22] through a fatigue law (Equation (2)) [23]. However, in overlays, the propagation phase is evaluated through fracture mechanics concepts, and the crack progression can be measured using Equation (1) [9].
N = a × (1/ε)^b^(2)
where N is the number of cycles; ε is the tensile strain (dimensionless); and a and b are the coefficients determined experimentally (dimensionless).

Numerical models to establish reflective cracking are performed by the crack simulation in the existing pavement, and the loading usually consists of Mode I [11,15]. Sousa et al. [24] proposed a method to evaluate cracking activity resulting from traffic loads, based on the determination of the layer’s modules, pavement layers’ thicknesses, air temperature, and the percentage of cracking surface. The rehabilitation design is expressed by von Mises strain to estimate the fatigue reflective cracking overlay lifespan.

Minhoto et al. [19] studied the reflective cracking using the finite element method and considered that the cracks initiation and propagation result from the overlay performance of degradation mechanisms. Such mechanisms depend on several factors such as traffic load, temperature variations, layer geometry, the properties of the materials of the layers, characteristics of the subgrade, existing cracks, and adhesion between layers.

Loria et al. [25] evaluated the resistance of HMA overlays using three analytical models. The findings showed that for the Virginia Tech Simplified Overlay Design Model, the overlay thickness was the major factor, which was followed by the thickness of the existing HMA layer. The Rubber Pavements Association Overlay Design Model had limited application for some HMA types. Other mixtures can be introduced in the method, but adjustment factors from site conditions are necessary.

In addition, the Mechanistic-Empirical Pavement Design Guide (MEPDG) of the American Association of State Highway and Transportation Officials (AASHTO) resulted in a constant overlay thickness to reach 100% reflected cracking after 20 years of lifespan regardless of the type of the HMA overlay performance or the pavement structure [25].

Delbono et al. [26] investigated the behaviour of reflective cracking using a geosynthetic material between each existing layer and the overlay. They concluded that geosynthetic reinforcement could delay the crack’s progression with effectiveness when it is located near the most stressed fibre of the overlay.

Other researchers addressed the reflective cracking, such as the development of correlations [27], use of non-destructive tests [28], through numerical simulation [29,30,31], by laboratory tests [32], and for airport pavements [33].

Worldwide research has proved that modified asphalt mixtures, such as asphalt rubber, enhance the resistance to fatigue, permanent deformation, thermal cracking, moisture damage resistance, and crack propagation retard in overlay [34,35,36,37,38,39,40,41].

Castillo et al. [42] used the Texas overlay test to evaluate the resistance of HMA to reflexive cracking. Four tested mixtures were modified, one with 50% tire rubber, one mix with 10% tire rubber, and 3% styrene block copolymer polymer. A mixture without modification was the reference. The author’s findings showed that the 50% crumb rubber mixture presented superior reflexive cracking performance, which is followed by the mixture with 10% crumb rubber. In addition, they concluded that applying asphalt rubber mixture (50% crumb rubber) is possible to achieve a relevant construction cost reduction (10% to 25% per lane-kilometre) with a practical level of consumption of scrap tires as well, extensive the span life and over long-term performance.

In Hong Kong, Xu et al. [43] developed research to improve the asphalt rubber performance with wastes polyethene terephthalate (PET) derived. They prepare additives from PET wastes chemically treated with triethylenetetramine (TETA) and ethanolamine (EA). Then, the additives were introduced into asphalt rubber (asphalt base PEN 60/70 and crumb rubber with a size of less than 30 mesh). In conclusion, it was demonstrated that waste PET additives increased the asphalt rubber performance. The asphalt rubber ageing resistance was studied by Li et al. [44]. The authors concluded that the crumb rubber helps mitigate asphalt oxidation, acting as a protective function.

One of the modified asphalt most applied in the Brazilian roads is asphalt rubber produced with crumb rubber powder from waste tires blended with asphalt [34]. The tire’s rubber is crushed from ambient or cryogenic methods [45,46]. Asphalt rubber can be produced by terminal blend (at refinery) or continuous blend (at asphalt plant) systems [34,45,46].

This study aims to assess the reflective cracking phenomenon through laboratory tests and numerical modelling to evaluate the overlay fatigue lifespan. Asphalt rubber mixtures were tested through the apparatus four-point bending and the Reflective Cracking Device (RCD), and a conventional mixture was produced as a reference. The numerical simulation was performed using the programme ANSYS 10.0 (Multiphysics) and the models developed by Minhoto [18] under the Brazilian temperature conditions.

## 2. Experimental Programme

### 2.1. Asphalt Mixtures

Two asphalt rubber mixtures with different granulometric curves and a reference mixture (conventional asphalt) were produced to perform the tests. The asphalt rubber mixtures followed the California Department of Transportation (asphalt rubber hot mixture gap-graded) [47] and the Asphalt Institute HMA dense-graded type IV [48] standards. The HMA grade “C” conventional mixture was designed under the Brazilian standard [49].

Asphalt rubber was made at the laboratory through the continuous blender system with a digestion time (mixing) of 90 min and 17% crumb rubber content. The asphalt base PEN 30/45 (classified by penetration) and two crumb rubbers (ambient and cryogenic) produced the modified asphalts.

The visual appearance of the crumb rubbers is shown in Figure 1. The morphology analysis was performed by scanning electron microscopy (SEM) with 50 times magnification (Figure 2). The cryogenic crumb rubber (Figure 2a) presented a uniform and regular grain structure, with a smooth texture. On the other hand, ambient crumb rubber (Figure 2b) had an irregular structure with different sizes and shapes. The researchers also observed a presence of agglomerates, in which the smaller particles are adhered to other, with a spongy appearance. These aspects are similar to those observed by Roberts et al. [50]. The laboratory production process for asphalt rubber is described and illustrated Appendix A.

The granite aggregates had the following gradations: grade 1 with particles size 6 to 12 mm; grade 2 with particles size 4 to 10 mm; grade 3 with particles size smaller than 4 mm. The mineral filler was also used to fit into gradation curves. All mixtures were designed through the Marshall Method, and Table 1 presents the mixture characteristics. Appendix A presents the flowchart of the experimental programme. The Appendix A also contain the phases of mixtures production (Appendix A), compaction (Appendix A), and the samples obtained process (Appendix A).

### 2.2. Four-Point Bending Modulus and Fatigue Tests

The dynamic modulus and fatigue laws were obtained at the laboratory using the flexural four-point bending test in controlled strain mode, according to the ASTM D3497 [51] and AASHTO TP 8 [52] standards, respectively.

The tests were performed using servo-hydraulic equipment (James Cox & Sons Inc., Colfax, CA, USA), consisting of a load structure, a hydraulic group, and a climatic chamber (Figure 3a). Inside is located the four-point bending device test (Figure 3b). The load structure consists of a vertical actuator connected to a servo valve, at the end of which is a load cell. The four-point bending device test is attached to the load frame below and above to the lower end of the vertical actuator. The climate chamber allows temperature control from −20 to +70 °C with an accuracy of ±0.5 °C, which is essential to maintaining a constant temperature throughout the test.

The dynamic modulus was measured in three temperatures (15, 20, and 25 °C) and seven frequencies established in the standard [51] (10, 5, 2, 1, 0.5, 0.2, and 0.1 Hz). The load cycles were 100 for the first three frequencies and 10 for the remaining frequencies. These tests were repeated for those three temperatures. The seven tests for each temperature were carried out on the same sample, and the reduced number of load cycles did not cause a significant reduction in the stiffness of the material. Then, the same sample was used to perform the fatigue tests.

Table 2 shows the asphalt mixtures’ dynamic modulus results. As expected, the dynamic modulus was lower for all mixtures for higher temperatures. At a reference temperature of 20 °C and frequency of 10 Hz, the conventional mixture (CONV) presented a higher modulus than the asphalt rubber ones, producing less stiffness asphalt.

The fatigue tests were carried out at 20 °C and 10 Hz. For each mixture, nine samples were tested at three strain levels, 200, 400, and 800 (10^−6^ mm/mm). The fatigue laws were described by Equation (1), and the mixture parameters are presented in Table 3. Considering a strain of 100 × 10^−6^ (N_100_) in Table 3, the dense-graded with ambient crumb rubber mixture (DGACR) had better fatigue performance, which is followed by the gap-graded mixture (GGCCR). The better fatigue resistance can be attributed to the elastic part provided by the tire rubber introduction. The conventional mixture (CONV) presented lower fatigue resistance.

### 2.3. Reflective Cracking Tests

The traffic action associated with the daily temperature variations leads to the crack edges being subjected to vertical and horizontal movements, and during the test, the crack activity is measured. This condition can be simulated by applying a two-dimensional stress state in the sample. The Reflective Cracking Device (RCD) developed by Sousa et al. [53] provides a method to perform the tests.

The RCD simulates the crack zone of the pavement overlay subjected to horizontal and vertical movements, leading to the crack propagation from the old layer to the overlay. Load-associated reflexive cracking is governed simultaneously by horizontal opening and (or) closing and a vertical shearing at the crack zone. The simulation of this process considers the simultaneity of these two modes of opening. The movements are measured by transducers placed on both sides of the sample.

The method used to assess the movements was the Crack Activity Meter (CAM) composed of two LVDT (Linear Variable Differential Transformers), one placed vertically and the other placed horizontally, allowing the measurement of both differential movements. The RCD is shown in Figure 4a, and the crack zone is represented in Figure 4b. A sample bonded in the equipment lower plate with an opening to simulate a crack is shown in Figure 5a, and Figure 5b presents the sample into the CAM. (RCD and CAN—James Cox & Sons Inc., Colfax, CA, USA).

The device is mounted with a superior plate. It is placed into the test equipment (Figure 6—James Cox & Sons Inc., Colfax, CA, USA), in which four pistons apply an effort to simulate the traffic load. The test configuration was established through a displacement application between 0.02 and 0.05 mm and forces between 200 and 400 N. The tests were performed at 20 °C and 10 Hz (the exact conditions of four-point bending tests) in the cylindrical sample (18 cm of diameter and 5 cm of thickness). The opening plate with 10 mm simulates a crack length with 2 to 3 mm in the field, and, for each mixture, six samples were tested. The test arrangement was developed by Sousa et al. [22] and Sousa et al. [53], which established the opening of the induced crack to represent the field.

### 2.4. Numerical Simulation

The numerical simulation was carried out using ANSYS 10.0 (Multiphysics) [54] and the model developed by Minhoto [18]. For the effect of the traffic action, the model was developed in a linear elastic regime, which considers the application of a vertical load simulating an axis of a vehicle and, for temperature variations, the viscoelastic regime was employed. Figure 7 shows the finite element model composed of two layers (asphalt mixture surface and granular base) representing the cracked pavement over the subgrade and the overlay. The model was generated using 8-node solid elements and three degrees of freedom per node (SOLID 185 element of ANSYS 10.0 Multiphysics software by ANSYS company, version 7.0, Canonsburg, PA, USA). The detailed description of the model is presented in the Appendix A.

The model geometry (Figure 5) comprises an overlay with 12 cm, an HMA cracked surface (21 cm), and a granular base layer (20 cm) over the subgrade (1.60 m). In the model, the old cracked layer and overlay interface is considered adhered. The adhered interface between these layers was recommended in the methodologies developed by Pais [14], Sousa et al. [22], and Sousa et al. [53]. Moreover, when promoting a connection between these layers, a monolithic system is formed, which is capable of withstanding the shear stresses applied by the traffic and temperature variations preventing possible slippage between them. Table 4 shows the dynamic modulus of the HMA cracked surface at five temperatures (frequency of 10 Hz) and Poisson’s coefficient. Table 5 presents the resilient modulus and Poisson’s coefficient for the granular base and subgrade layers.

To take into account the temperature variations during the pavement life requires that the modulus be shown as a function of temperature. For temperatures above 25 °C, the modulus variation is calculated using Equation (3), and for temperatures lower than 25 °C, the modulus variation is calculated using Equation (4). Table 6 shows the results.
(3)logE = a + T × b
(4)E = a + T × b
where E is the dynamic modulus (MPa); T is the temperature (°C); a and b are experimental coefficients.

In the simulation of pavement behaviour under temperature variations, the asphalt mixture’s viscoelastic behaviour has to be expressed by the sheer volumetric relaxation modulus calculated using a Prony series. In this way, the finite element model application requires a fatigue law of the overlay as a function of the temperature. The fatigue parameters laws obtained at the laboratory (Table 3) were shown as a function of tensile strain and obtained at a temperature of 20 °C and, therefore, cannot be used directly in the finite element model. In this way, the asphalt mixtures fatigue laws (Equation (5)) are expressed in the strain and the dynamic modulus (Table 7).
N = a × E^b^ × ε^c^(5)
where N is fatigue cycles, E is the dynamic modulus (MPa), ε is the tensile strain; and a, b, and c are experimental coefficients.

The damage related to reflective cracking was obtained using Equation (6). Appendix A presents the annual traffic per axle.
D = N/T(6)
where D is the damage due to traffic, N is the overlay predicted life, and T is the traffic (80 kN equivalent single axle load—ESAL).

From the temperature profile, it is possible to obtain the response of pavement structure with the overlay relative to von Mises strains (Equation (7)), over the 24 h, by simulating the temperature variations action. Everyday repetition over the month allows obtaining response behaviour throughout the month. The analysis and the simulations over the 12 months (year) were also performed to measure the annual overlay damage.
ε_VM_ = [½(ε_1_ − ε_2_)^2^ + (ε_1_ − ε_3_)^2^ + (ε_2_ − ε_3_)^2^]^0^^.^^5^(7)
where ε_VM_ is the von Mises strains; ε_1_, ε_2_, and ε_3_ are the principal strains.

The air temperature over a month has slight variations when a year is considered. In this case, the overlay behaviour analysis can be accomplished for a representative month–temperature profile. The maximum and minimum temperatures are the maximum and minimum average daily ones throughout the month. A similar analysis was taken into account for the months. Thus, the annual damage comprising temperature variations was calculated using Equation (8).
D = a × T^b^_max_ × e^c^^x^^Δt^(8)
where D is damaged due to temperature variations; T_max_ is the maximum air daily temperature (°C); Δt is the air daily amplitude temperature (°C); e is the Neperian number; a, b, and c are experimental coefficients.

## 3. Results

### 3.1. Damage Evaluation

The monthly damage was obtained for each of the three mixtures through fatigue laws (Table 7). The numerical analysis results for the five temperature conditions (maximum and amplitude) allowed an estimate of the monthly damage (Table 8). Using Equation (8) and Table 9, the coefficients presented in Table 9 were obtained.

Minhoto [18] studied Portugal (North hemisphere), where the warmer months occur between May and September. However, Brazil is located in the southern hemisphere, and the warm conditions occur between December and March. Therefore, for south Brazil conditions, the parameters must be calculated for a maximum temperature of 25 °C and thermal amplitude of 15 °C. Appendix A presents the prediction of the average monthly damage of each mixture, adapted to the southern Brazil climatic conditions. The sum of monthly damage (Appendix A) results in the annual damage (Figure 8), revealing that the most severe damage was observed for the conventional mixture (CONV).

### 3.2. Reflective Cracking Laboratory Tests

The laboratory tests (RCD) relate the load cycles for which mixtures fail for reflective cracking. The failure is measured at the load cycle in which a crack in the sample develops a 1 mm opening. The test frequency was 10 Hz, and the test temperature was 20 °C. Figure 9 illustrates the sample condition in which the bottom–up crack was visible after the test.

The crack opening that develops in the samples during the RCD test presents an evolution in the number of load cycles (Figure 8, results of a DGACR sample). A 4th-degree polynomial approximation was obtained for each sample, as represented by Equation (9). In Figure 8, the shown equation represents the results of the DGACR mixture, as shown in sample 4. Appendix A presents the tests results of the mixtures samples.
Load Cycles = a × (c)^4^ + b × (c)^3^ + c × (c)^2^ + d × (c) + e(9)
where c is the crack length (mm); a, b, c, d, and e are experimental coefficients.

It is possible to distinguish three regions in Figure 10. Region I is characterised by crack initiation. In Region II, it is noted that the crack propagates at a slower speed. Finally, in Region III, the cracking increases until the failure happens. The curve regions from the RCD test are similar to those observed by Paris and Erdogan [9], as shown in Appendix A.

### 3.3. Von Mises Strain Related from RCD Tests

A numerical analysis was carried out to obtain the von Mises strain performed using ANSYS 10.0 (Multiphysics). Figure 11 shows the finite element mesh, and Figure 12 presents the von Mises tension state.

The Appendix A for the horizontal and vertical displacements and strain generated in the sample. Appendix A represents the sample distortion and von Mises strain. The tension state to which the test samples were subjected is shown in Appendix A.

Equation (10) represents the overlay resistance of reflective cracking from von Mises strains under the conditions the pavement is submitted. The reflective cracking fatigue curves are shown in Figure 13. The laws show reflective cracking fatigue life to any value of the von Mises strain. The von Mises strain results are presented in Appendix A.
N_RC_ = a × (1/ε_VM_)^−^^b^(10)
where N_RC_ is the fatigue resistance to reflective cracking; ε_VM_ is the von Mises strain (10^−^^6^); a and b are experimental coefficients.

The results confirmed that among all mixtures submitted to the same von Mises strain level in the pavement structure, GGCCR presented a higher reflective cracking fatigue performance.

## 4. Discussion

Dynamic modulus tests are carried out to evaluate the HMA stiffness relative to the temperature and frequency variations. In Brazil, the modulus measured at 20 °C and 10 Hz is used to characterise the HMA surface layer. Fatigue tests are accomplished to obtain the curve parameters, and the pavement structure can be designed. However, it is also essential to predict the HMA performance in the rehabilitation design to ensure the expected lifespan.

The overlay is subject to reflective cracking, which is not always adequately provided for in rehabilitation projects. Test and numerical simulations can consider reflective cracking in rehabilitation design. In this work, procedures for evaluating reflective cracking in overlays were presented using laboratory tests and numerical simulation.

Table 10 summarises the conventional and asphalt rubber mixtures related to reflective cracking performance. The overlay life, in years, was calculated as the inverse of the annual damage. In relative terms, an overlay lifetime, for damage, of an asphalt rubber would be almost six times greater than a conventional one. As for overlay damage under Brazilian temperatures conditions, a dense-graded asphalt rubber mixture (DGACR) can be proper for overlays.

Both asphalt rubber mixtures were more resistant to reflective cracking than to HMA. The asphalt rubber mixture GGGCR presented better performance to reflective cracking. In addition to the improvements due to asphalt rubber use, the gap-graded granulometry showed a suitable alternative to overlays in terms of reflective cracking.

An important finding to be pointed out is that the fatigue tests (for example, four-point bending) results are commonly used to overlay design without taking into account the reflective cracking evaluation. However, the relative life analysis of the mixtures indicated that the fatigue performance obtained in the four-point bending tests does not rebound in the reflective cracking fatigue life. The relative life of 57.7 times is reduced to 7.9 times related to the laboratory tests. Such a finding reveals the importance of assessing reflective cracking in HMA, minimising mistakes in the overlay design and improving rehabilitation life.

The validation of the numerical simulation and RCD tests related to reflexive cracking fatigue life was obtained by correlation, as shown in the example given in Figure 14 for the conventional mixture. The results show that the RCD tests results and numerical simulation correlated well, indicating that the analysis was suitable.

## 5. Conclusions

In this work, a comprehensive reflective cracking analysis was accomplished to evaluate the asphalt rubber mixture’s performance as an overlay. A numerical simulation and laboratory tests (modulus, fatigue and RCD) were fulfilled. The reflective cracking damage due to traffic and temperature variations was also estimated.

The numerical simulation was performed through a finite element model from data measured in RCD tests. It was possible to obtain reflective cracking fatigue law.

Both asphalt rubber mixtures obtained more resistance to reflective cracking than to HMA. In relative terms, an overlay lifetime, for damage, of an asphalt rubber would be almost six times greater than a conventional one.

The relative life analysis of asphalt rubber mixtures observed in the four-point bending fatigue tests was at least 57.7 times higher than the conventional HMA. However, the relationship is reduced to 7.9 times related to the fatigue reflective cracking tests. Such a finding reveals the importance of assessing reflective cracking in HMA, minimising mistakes in the overlay design and improving rehabilitation lifespan.

## Figures and Tables

**Figure 1 materials-15-02375-f001:**
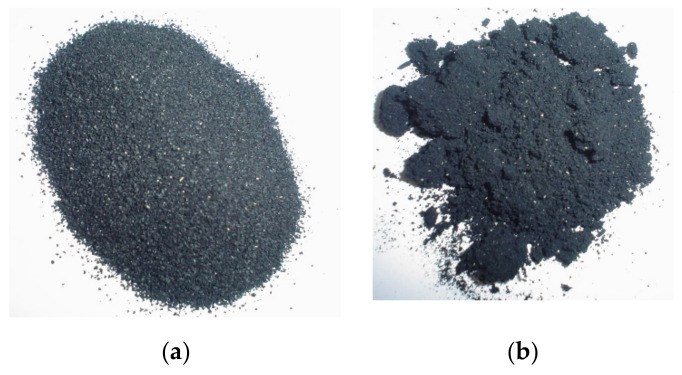
Crumb rubbers visual appearance. (**a**) Cryogenic crumb rubber. (**b**) Ambient crumb rubber.

**Figure 2 materials-15-02375-f002:**
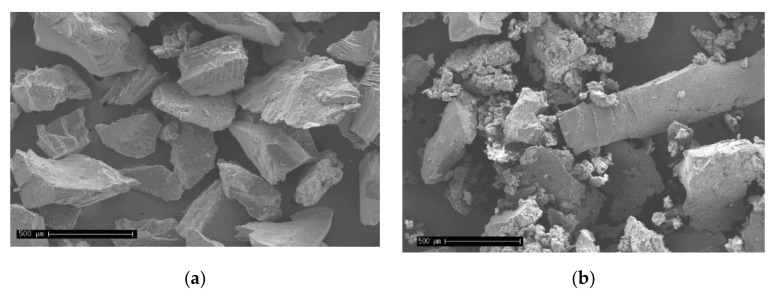
Morphology aspect by SEM (50 times magnification). (**a**) Cryogenic crumb rubber. (**b**) Ambient crumb rubber.

**Figure 3 materials-15-02375-f003:**
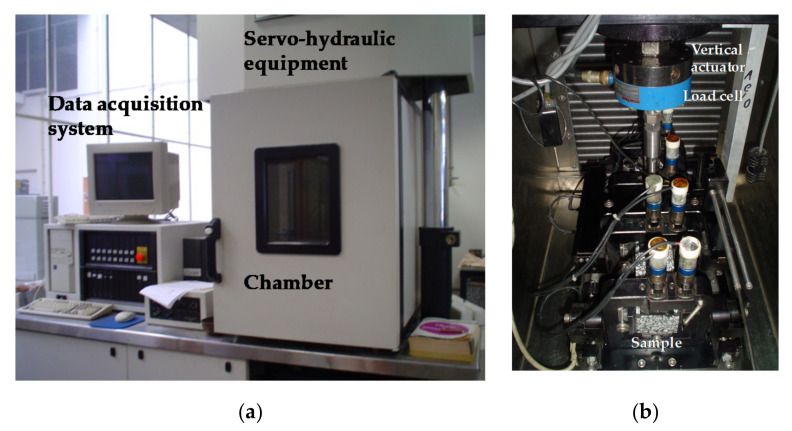
Servo-hydraulic equipment and four-point bending device test. (**a**) Servo-hydraulic equipment. (**b**) Four-point bending device test.

**Figure 4 materials-15-02375-f004:**
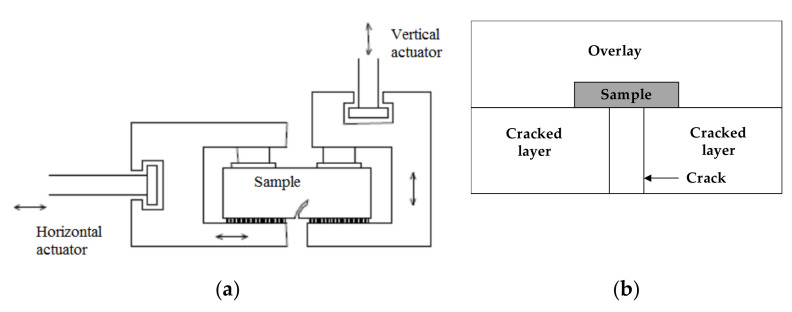
Schematic representation of the RCD and the crack zone. (**a**) RCD device. (**b**) Crack zone.

**Figure 5 materials-15-02375-f005:**
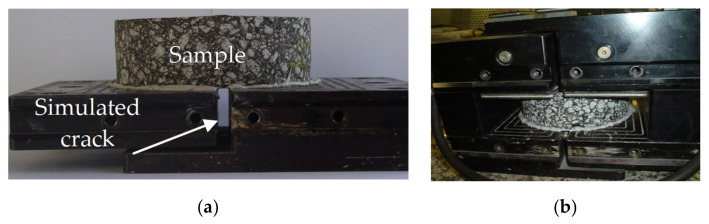
Samples, simulated crack, and the assembled set. (**a**) Sample and simulated crack RCD device. (**b**) Sample into the CAM.

**Figure 6 materials-15-02375-f006:**
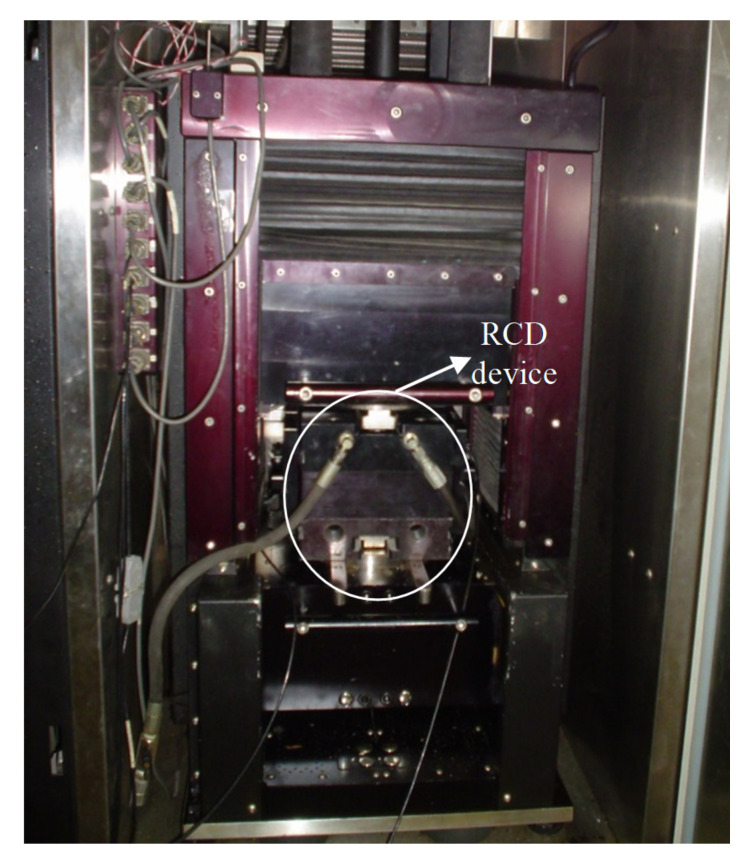
RCD device mounted into the test equipment.

**Figure 7 materials-15-02375-f007:**
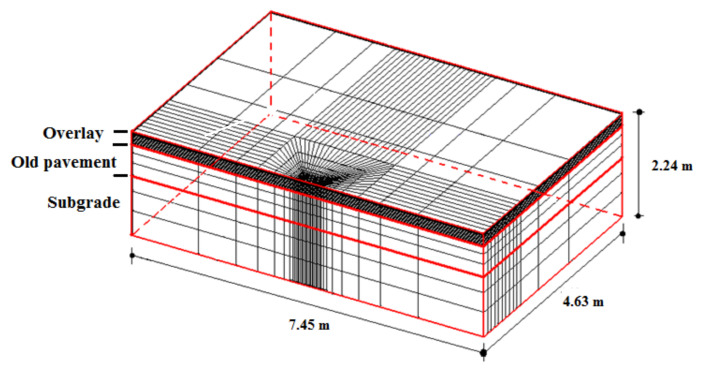
The finite element model (based on [18]).

**Figure 8 materials-15-02375-f008:**
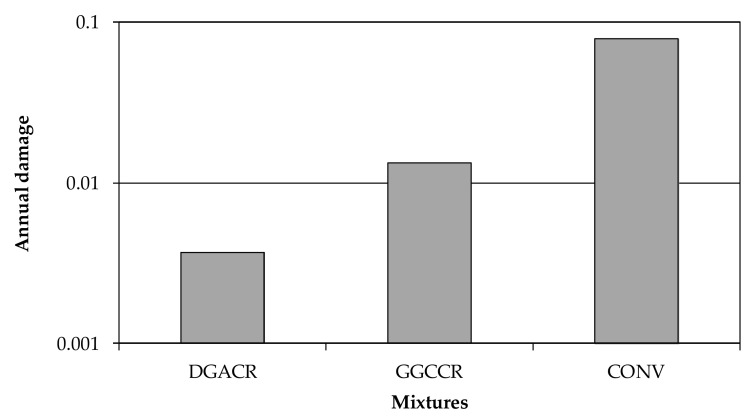
Annual damage for the mixtures as an overlay of 12 cm.

**Figure 9 materials-15-02375-f009:**
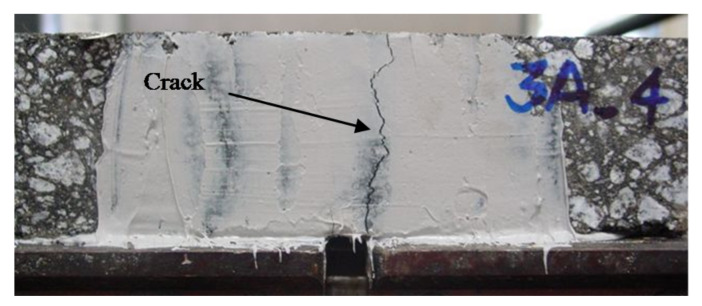
Reflective crack example produced on the sample after the test (side view).

**Figure 10 materials-15-02375-f010:**
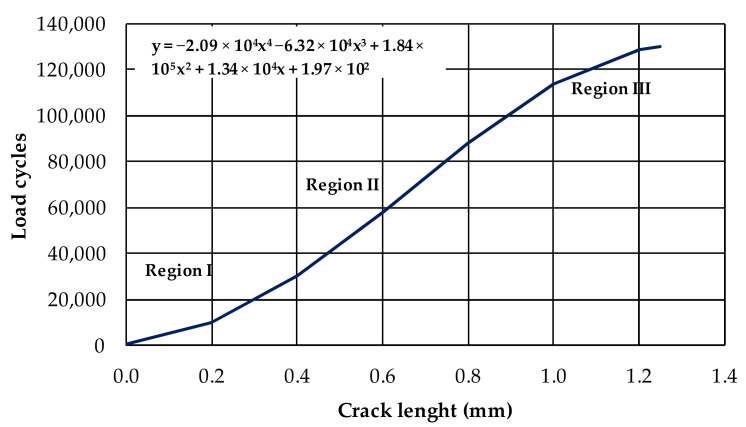
Crack opening concerning the load cycles in RCD tests.

**Figure 11 materials-15-02375-f011:**
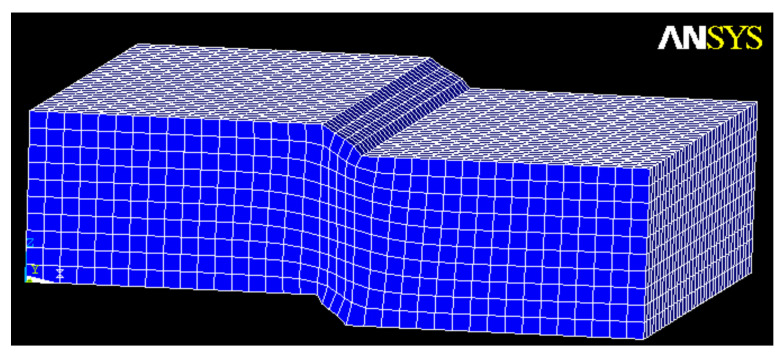
Finite element mesh (deformed).

**Figure 12 materials-15-02375-f012:**
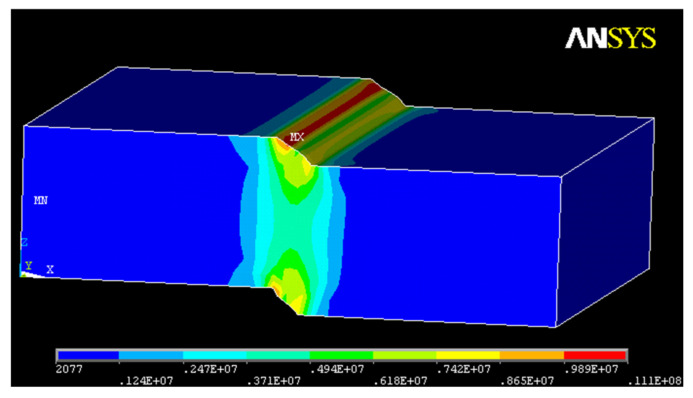
Von Mises’s tension state.

**Figure 13 materials-15-02375-f013:**
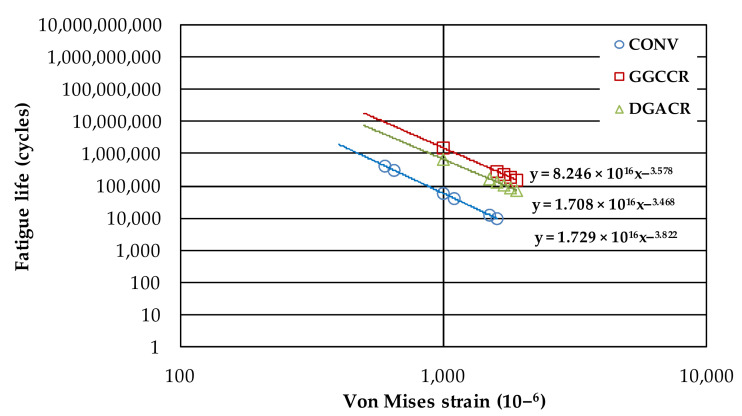
Reflective cracking fatigue curves.

**Figure 14 materials-15-02375-f014:**
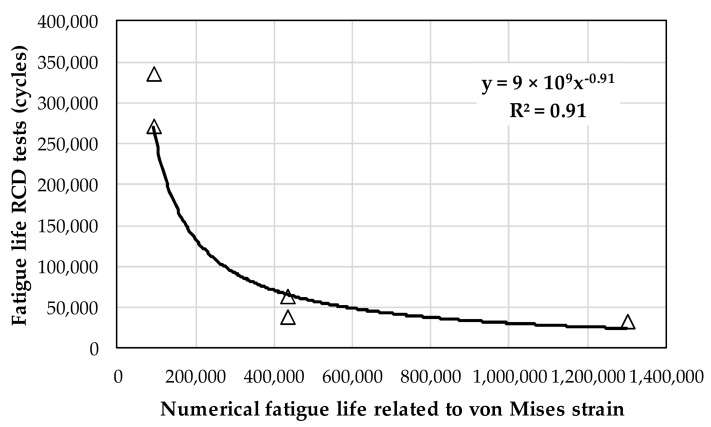
Example of correlation between RCD tests and numerical simulation.

**Table 1 materials-15-02375-t001:** Asphalt mixtures characterisation.

Mixture	Asphalt Base	Crumb Rubber	Rubber Content	Gradation Type	Voids Content	Asphalt Content
GGGCR	PEN 30/45 ^1^	Cryogenic	17%	Gap-graded [47]	6.0%	8.0%
DGACR	PEN 30/45 ^1^	Ambient	17%	Dense-graded [48]	5.0%	7.0%
CONV	PEN 50/7 0 ^2^	-	-	Dense-graded [49]	4.0%	5.5%

PEN 30/45 ^1^—Penetration 30–45 (0.1 mm); PEN 50/70 ^2^ Penetration—50–70 (0.1 mm), both classified by penetration grade at 25 °C, 100 g.

**Table 2 materials-15-02375-t002:** Dynamic modulus of the mixtures (20 °C, 10 Hz).

Temperature (°C)	Mixtures and Dynamic Modulus (MPa)		
	GGGCR	DGACR	CONV
15	6516	7344	8516
20	5174	6132	6451
25	3833	4921	4387

**Table 3 materials-15-02375-t003:** Asphalt mixtures fatigue parameters and N_100_.

Mixtures	Fatigue Parameters and Cycles for 100 × (10^−6^ mm/mm) Strain			
	a	b	R^2^	N_100_ (Cycles)
GGGCR	2.782 × 10^17^	4.597	0.96	1.78 × 10^8^
DGACR	4.852 × 10^19^	5.463	0.99	5.74 × 10^8^
CONV	1.185 × 10^15^	4.037	0.99	9.99 × 10^6^

**Table 4 materials-15-02375-t004:** HMA cracked surface parameters (based on [18]).

Temperature (°C)	Dynamic Modulus (MPa)	Poisson’s Coefficient
−5	12,000	0.35
0	9000	0.35
5	6500	0.35
10	4000	0.35
15	2500	0.35
25	680	0.35

**Table 5 materials-15-02375-t005:** Base and subgrade parameters (based on [18]).

Layer	Resilient Modulus (MPa)	Poisson’s Coefficient
Granular base	270	0.40
Subgrade	90	0.45

**Table 6 materials-15-02375-t006:** Cracked asphalt mixture surface modulus calculated using Equations (3) and (4).

Temperature (°C)	Mixtures and Dynamic Modulus (MPa)		
	GGGCR	DGACR	CONV
−5	11,881	12,188	16,773
5	9199	9788	12,644
15	6516	7344	8516
20	5174	6132	6451
25	3833	4921	4387
40	1746	2691	1678
50	1026	1795	871

**Table 7 materials-15-02375-t007:** Fatigue law parameters (Equation (5)) taking the temperature into account.

Coefficients	Mixtures		
	GGGCR	DGACR	CONV
a	−2.1 × 10^−3^	−9.32	−3.3 × 10^−4^
b	−0.807	−1.523	−0.632
c	−4.695	−5.684	−4.191

**Table 8 materials-15-02375-t008:** Monthly damage.

T_max_^1^ (°C)	ΔT^2^ (°C)	Mixtures		
		GGGCR	DGACR	CONV
25	25	4.793 × 10^−4^	1.088 × 10^−4^	3.912 × 10^−3^
35	10	3.528 × 10^−2^	1.067 × 10^−2^	1.150 × 10^−1^
15	15	1.440 × 10^−4^	3.453 × 10^−5^	1.249 × 10^−3^
10	10	8.775 × 10^−5^	2.167 × 10^−5^	6.592 × 10^−4^
25	15	7.068 × 10^−4^	1.441 × 10^−4^	5.108 × 10^−3^

T_max_^1^—maximum temperature; ΔT^2^—thermal amplitude.

**Table 9 materials-15-02375-t009:** Parameters for the monthly damage (Equation (8)).

Coefficients	Mixtures		
	GGGCR	DGACR	CONV
a	1.94 × 10^−8^	4.05 × 10^−8^	2.17 × 10^−7^
b	4.37	3.96	3.97
c	−0.177	−0.232	−0.130

**Table 10 materials-15-02375-t010:** Summary of the results of the tests.

Test/Parameter	Mixtures		
	GGGCR	DGACR	CONV
Damage			
Life (years)	74.9	268.5	12.8
Relative life	5.9	21.0	1.0
Four-point bending			
ε (10^−6^) ^1^	100	100	100
N_100_ (cycles)	1.78 × 10^8^	5.75 × 10^8^	9.99 × 10^6^
Relative life	17.8	57.7	1.0
Numerical simulation			
ε_VM_ (10^−6^) ^2^	154	128	141
Fatigue life (cycles)	1.23 × 10^9^	8.41 × 10^8^	1.06 × 10^8^
Relative life	11.6	7.9	1.0

^1^ ε—tensile strain; ^2^ ε_VM_—von Mises strain.

## Data Availability

Not applicable.

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
