# Peer review of "Performance of Asphalt Rubber Mixture Overlays to Mitigate Reflective Cracking"

_materials, 2022, doi:10.3390/ma15072375_

Round 1

Reviewer 1 Report

The research program was well planned and the research findings were presented and discussed properly. It adds valuable remarks for the pavement overlay design. Therefore it can be accepted for publication after carrying on minor revision. Here are some recommendations for improvement: 1) The three figures in Introduction are not needed; 2) The interface definition between overlay and old pavement layer in their model should be discussed; 3) The indication of tension scale in Fig 10 is not clear for getting detailed data. The data displayed at the right-up corner on the Fig 9 and 10 is Jan 2 2009, did the author carry on such simulation 13 years ago?

Author Response

We are very grateful for your comments and suggestions. We have made improvements in the paper to consider your suggestions. Our responses to your comments are in the attached document.

Reviewer 2 Report

Line 11 says: This failure occurs of cracks in the lower layers that propagate to the overlay 11

due to traffic loads and temperature variations. Comment: Could it happen because of the soil foundation layers?

Line 74: Figure with reflective cracks. Comment: maybe crack lines could be thicker

Line 78. Here it is noticed a phrase not included in the abstract, exact! it is not indicated in the abstract!

Line 163: Table 1. Asphalt mixtures characterisation. Comment: Are these asphalt mixtures related to the rubber size?

Line 170 says: The dynamic modulus was measured in three temperatures (15, 20 and 25oC), Comment: Why? Lab conditions? Are these three temperatures common in Brazil? Or in Portugal?

Line 176: Table 2. Dynamic modulus of the mixtures. Comment: Which of the seven frequencies (10, 5, 2, 1, 0.5, 0.2 and 0.1 Hz). It it not enough clear.

Line 201: cylindrical samples? If the sample is cylindrical why did you write 18 cm x 18 cm x 5 cm? 18cm diameter and 5 cm thickness?

Lines 216 and 217 says: The model geometry (Figure 5) comprises an overlay with 12 cm; an HMA cracked surface (21 cm), a granular base layer (20 cm) over the subgrade (1.60 m). Comment: the add is: 12 + 21 + 20 + 160 = 213 cm = 2.13 m Table 4 says 2.24 m, and it is different of 2.13m

Lines 221 and 222: Tables 4 and 5, say Modulus, does it mean Resilient Modulus?

In the filed attached in PDF format, We wrote comments, please look at it

Author Response

We are very grateful for your encouraging comments and suggestions. Our responses to the comments are presented in the attached document.

Reviewer 3 Report

This case study mainly evaluated the reflective cracking behavior of asphalt rubber mixture as overlay through laboratory test and numerical simulation, for the purpose of prolonging the service life of asphalt pavement. The topic is good but some shortcomings are still existent. Therefore, major revisions are given with the comments below.

- The Introduction part should make the current research clear for better understandings with careful improvement. As well, it still needs to introduce some performance characteristics of asphalt rubber through citing relevant references. The recent literatures related to asphalt rubber are below for your reference:

1 Evaluation of cracking resistance of tire rubber–modified asphalt mixtures. Journal of Transportation Engineering, Part B: Pavements, 2021, 147(3), p.04021019.

2 Sustainable practice in pavement engineering through value-added collective recycling of waste plastic and waste tyre rubber. Engineering, 2021, 7(6), pp.857-867.

3 Effects of rubber absorption on the aging resistance of hot and warm asphalt rubber binders prepared with waste tire rubber. Journal of Cleaner Production, 2021, 303, p.127082.

- Line 200-203: The force application mode and working mode of RCD device shall be more described in detail. Please further elaborate on why “the opening plate with 10 mm simulates a crack length with 2 to 3 mm in the field”.

- The correlation between numerical simulation and experimental results should be confirmed for the validation of the findings.

- The manuscript needs be proofread further.

Author Response

(The authors gave the same response as above.)

Reviewer 4 Report

In this research, behaviour of rubber mixture asphalt as an overlay is evaluated numerically and experimentally by investigating its reflective cracking. Parameters such as dynamic modulus, fatigue are presented and compared.

The subject matter is of interest and the manuscript is easy to follow and understand. The following comments, however, should be addressed by the authors in order to enhance the article quality:

  1. Section 2.1: the manuscript suffers from adequate illustrations of both numerical and experimental study. As an example, the lab photo representing the mixture should be added to the manuscript. The shape and geometry of the rubbers should be also displayed and explained.
  2. Section 2.2: The testing process is not clearly explained as well: the test process for measuring dynamic modulus and fatigue should be explained by the authors. More importantly, the lab photo representing test setup should be added.
  3. Page 7: the reference of the equations used in the study (e.g., Eq. 3 and 4) should be mentioned.
  4. Section 3.3: validation of the numerical model is not presented appropriately in the manuscript. Verification is a critical part of a numerical study which shows the reliability of a model. The authors are therefore recommended to verify the FEA model by experimental results (e.g., by comparing stress-strain curve and failure of model and sample).

Author Response

(The authors gave the same response as above.)

Reviewer 5 Report

1. In the table 2 there is not the frequency at which the dynamic modulus was determined.
2. page 8 line 262 - 3 mixtures, not 5.

Author Response

(The authors gave the same response as above.)

Round 2

Reviewer 3 Report

All of the responses went up to my comments. This version of the manuscript is well revised and satifactory for acceptance.